# Knowledge and practice of facemask disposal among university students in Thailand: A new normal post the COVID-19 pandemic

Narisara Kaewchutima[1]*, Nopadol Precha[1,2], Netnapa Duangkong[1], Anthika Jitbanjong[1], Ni Made Utami Dwipayanti[3]

1 Department of Environmental Health and Technology, School of Public Health, Walailak University, Nakhon Si Thammarat, Thailand, 2 One Health Research Center, Walailak University, Nakhon Si Thammarat, Thailand, 3 School of Public Health, Faculty of Medicine and Health Sciences, Udayana University, Bali, Indonesia

* narisara.ka@mail.wu.ac.th

## Abstract

The use of facemasks is essential to prevent the transmission of COVID-19. University students are a significant demographic that generates substantial infectious waste due to the new normal practice of using disposable facemasks. In this cross-sectional study, we investigated the facemask disposal knowledge and practices among university students in Thailand between September and October 2022. We used a self-report questionnaire comprising 29 questions to determine the students' demographic characteristics and facemask disposal knowledge and practices. We then applied a logistic regression model to estimate the association between the students' facemask disposal knowledge and practices and their demographic characteristics. A total of 433 participants completed the questionnaire comprising health science (45.27%) and non-health science (54.73%) students. Surgical masks were the most popular masks (89.84%), followed by N95 (26.33%) and cloth masks (9.94%). While their levels of knowledge regarding facemask disposal were poor, the students' practices were good. The factors associated with proper facemask disposal were sex (AOR = 0.469, 95% CI: 0.267, 0.825), academic grade (AOR = 0.427, 95% CI: 0.193, 0.948), and knowledge level (AOR = 0.594, 95% CI: 0.399, 0.886). No demographic factors influenced knowledge. Our findings highlight the influence of facemask disposal knowledge on students' disposal practices. Information promoting the appropriate disposal practices should therefore be promoted extensively. Furthermore, continuous reinforcement by raising awareness and educating students on proper facemask disposal combined with the provision of adequate infectious waste disposal facilities could help reduce the environmental contamination of infectious waste and thus improve general waste management.

**Data Availability Statement:** All relevant data are within the manuscript.

**Funding:** The authors received no specific funding for this work.

**Competing interests:** The authors have declared that no competing interests exist.

## Introduction

The emergence of the novel coronavirus disease (COVID-19) has dominated public attention globally since December 2019 [1, 2]. The disease is highly contagious and can be transmitted through respiration and contact with contaminated droplets that are released when people speak, cough, and sneeze [3, 4]. The methods for the prevention and control of COVID-19 infection include regular hand hygiene, maintaining physical distancing, and the use of masks [5–7]. Evidence supports the use of facemasks to significantly reduce the risk of virus transmission [8–10].

The use of different types of personal protective equipment (PPE), including facemasks, face shields, gloves, gowns, and head mobs, were required in public places, hospitals, and schools (i.e., high schools, colleges, and universities) to control transmission during the COVID-19 pandemic. A consequence of the extensive use of PPE has been the enormous demand for single-use plastic products, which have generated enormous volumes of infectious waste. In addition, increased municipal solid waste and medical waste have been reported in different regions [11–16]. Although the pandemic has passed, infections are continuing. Facemasks are therefore required for individuals who appear healthy due to the continued prevalence of asymptomatic infections [17]. The resultant increased production and consumption of plastic medical products poses a considerable challenge to waste management [5, 18, 19]. Furthermore, this waste may have severe adverse impacts on human and environmental health [11, 20–22].

Facemasks are no longer mandatory following the pandemic, but their use has become the new normal among the general public. University students are part of a large community that produces massive volumes of infectious waste daily. The effective management of used facemasks is thus necessary to prevent the reemergence of infectious diseases and the emergence of new contaminants. Proper waste management in the community should be encouraged, but this is significantly influenced by community knowledge, attitudes, and practices [15, 23, 24]. Studies on knowledge, attitudes, and practices could provide baseline information to guide interventions aimed at improving perceptions and practices regarding facemask disposal. Accordingly, in this study we aimed to determine the knowledge and practices relating to facemask disposal and the associated factors among university students in Thailand. Our findings could help improve the current management of infectious waste at universities and thereby enhance waste management sustainability.

## Methodology

### Study design and setting

We conducted this cross-sectional study at Walailak University in Nakhon Si Thammarat, Thailand, between September and October 2022. Walailak University is a public university, which had 8,205 students enrolled at the time of the study. It comprises 14 schools, three international colleges and one graduate college and includes a wide range of health science, science and technology, and social science programs.

### Ethical accreditation

The study protocol was approved by the Human Research Ethics Committee of Walailak University (WU; Reference number: WUEC-22-249-01). All the volunteers provided their informed consent to participate in the study. The objectives and other important information regarding the study were explained to them. The confidentiality of the research findings was

strictly maintained, and the study was performed in line with the relevant guidelines and regulations.

## Sample size

A sample size of 368 subjects was calculated using the Krejcie and Morgan equation:

$$n = \frac{Np(1-p)z_{1-\frac{\alpha}{2}}^2}{d^2(N-1) + p(1-p)z_{1-\frac{\alpha}{2}}^2}$$

The sample size was based on a reference population of the 8,205 undergraduate students at Walailak University at a 95% confidence level, a 5% margin of error, and a population ratio of 0.5. We anticipated dropout data, so sampling corrections of 20% were included in the sample size calculation. In total, 442 students were recruited for our study. They comprised 242 non-health science (NHS) students and 200 health science (HS) students, which reflected the proportion of students in these areas of study at the university.

## Sampling method

The 8,205 undergraduate students at the university were considered the source population, and simple random sampling was applied. We included undergraduates aged at least 18 years who were enrolled in their first to fourth years of study and who provided their informed consent to participate in the study. We excluded students who were not enrolled as undergraduates, were younger than 18, did not sign the consent form, or provided incomplete answers to our questionnaire.

## Questionnaire design and quality assurance procedures

The questionnaire included general questions regarding the students' demographic information, four questions on their facemask use, one question on their sources of information regarding facemask disposal, and 20 questions on their facemask disposal knowledge and practices (10 questions for each of knowledge and practices). The questionnaire was developed after reviewing the literature on waste management [13, 25–27]. All the questions were reviewed and evaluated by experts from the fields of public health, waste management, and environmental engineering to determine their content, relevance, and readability. The questionnaire was modified until it scored a content validity index of 0.70 or higher. The questionnaire used in this study had a content validity index of 0.87, which was acceptable [28, 29]. A pilot study with 30 students from another university was conducted to improve the final version of the questionnaire. The participants in the preliminary study were excluded from the final analysis. The reliability of the questionnaire was measured using Cronbach's alpha, and the value obtained indicated that the questionnaire had a reliability level of 0.80. The questionnaire was prepared in English and translated into Thai before the data were collected.

In the questions that evaluated the respondents' knowledge of facemask disposal, we used the categorical responses "yes" or "no". Each correct answer scored one point, and each incorrect answer was given a score of zero. The questions regarding proper practices were graded with 0 points for "never," 1 for "sometimes," and 2 for "always," while the questions about improper practices were reverse graded.

## Data collection

Before beginning the survey, all the participants were given information about the study, including the title, objectives, and knowledge and practice questions. They were also provided

with a consent form, and the participants who decided to take part in our study were asked to sign the form. Each participant spent approximately 30 minutes answering the questions. The principal investigator examined the obtained data for completeness, accuracy, clarity, and consistency. The questionnaires with incomplete responses were deemed invalid.

## Data analysis

SPSS version 25 (IBM) was used to manage and analyze the data collected from the completed questionnaires. The descriptive statistics and Fisher's exact test were used to analyze the qualitative variables and compare the characteristics of the HS and NHS students. In contrast, the quantitative variables were summarized using the mean ± standard deviation (SD). The respondents who scored higher than the median were assessed as having "good knowledge" and "good practices" relating to facemask disposal [23, 30, 31]. The associations between the categorical variables were analyzed using chi-square tests. Spearman's correlation test was used to estimate the strength and direction of the relationship between the students' knowledge and practices and the demographic variables (i.e., sex, academic grades, and academic program). A logistic regression model was applied to estimate the association between the students' knowledge and practice levels regarding facemask disposal and their demographic characteristics. Adjusted odds ratios (AOR) and 95% confidence intervals (CIs) were used to indicate the existence and strength of the correlation between the dependent and independent factors. The significance level was set at 0.05 for all the statistical tests.

## Results

### Demographic characteristics and facemask use

A total of 433 (97.96%) participants completed the questionnaire. They were divided into two groups based on their major programs: 45.27% (n = 196) were majoring in the following health science programs: Medicine, Pharmacy, Nursing, Allied Health, and Public Health. The remaining 237 (54.73%) students were enrolled in NHS programs (Art, Science, Political Science and Law, Management, Informatics, Agricultural Technology and Food, Engineering and Technology, and Architecture and Design). Our study included more female than male students and more freshmen than seniors (Table 1). In terms of facemask usage behaviors, most of the participants used two disposable facemasks daily, while a few used three or more. Up to 70% of all the participants, particularly the HS students, did not reuse their facemasks. The students frequently wore their facemasks for over three hours: more than half the NHS students wore them for three to five hours, and the HS students wore them for longer than five hours. The most popular type of mask used by the students was surgical masks, followed by N95 and cloth masks (Fig 1). All types of facemasks were used repeatedly, including surgical masks (28.00%), N95 marks (39.44%), and cotton masks (37.21%).

More than 65% of the students received information on facemask disposal via Facebook, while the university board and Line were lower perception channels (Fig 2). Our further findings revealed that the students' academic programs were associated with their selection of surgical facemasks and visits to government agency websites for facemask disposal information ($p < 0.05$).

### Knowledge of facemask disposal

Overall, the students had poor knowledge with respect to facemask disposal, as the average score was 6.78 ± 1.77. The knowledge levels of the HS students were good, but the NHS students had inadequate knowledge of facemask disposal. These findings suggest that the HS

**Table 1. Demographic characteristics and facemask use among the students ($N$ = 433).**

| Variables | n (%) Total (N = 433) | n (%) NHS (N = 237) | n (%) HS (N = 196) | P (Fisher's Test) |
|---|---|---|---|---|
| **Sex** | | | | |
| Male | 68 (15.70%) | 50 (21.10%) | 18 (9.18%) | 0.001 |
| Female | 365 (84.30%) | 187 (78.90%) | 178 (90.82%) | |
| **Academic grade** | | | | |
| 1st year | 252 (58.20%) | 158 (66.67%) | 94 (47.96%) | <0.001 |
| 2nd year | 80 (18.48%) | 52 (21.94%) | 28 (14.29%) | |
| 3rd year | 64 (14.78%) | 21 (8.86%) | 43 (21.94%) | |
| 4th year | 37 (8.55%) | 6 (2.53%) | 31 (15.82%) | |
| **Income** | | | | |
| <3000 | 117 (27.02%) | 73 (30.80%) | 44 (22.45%) | 0.324 |
| 3000–5000 | 214 (49.42%) | 112 (47.26%) | 102 (52.04%) | |
| 5001–10000 | 97 (22.40%) | 49 (20.68%) | 48 (24.49%) | |
| 10001–15000 | 2 (0.46%) | 1 (0.42%) | 1 (0.51%) | |
| >15000 | 3 (0.69%) | 2 (0.84%) | 1 (0.51%) | |
| **Number of facemasks used per day** | | | | |
| 1 | 113 (26.10%) | 70 (29.54%) | 43 (21.94%) | 0.249 |
| 2 | 248 (57.27%) | 126 (53.16%) | 122 (62.24%) | |
| 3 | 50 (11.55%) | 28 (11.81%) | 22 (11.22%) | |
| >3 | 22 (5.08%) | 13 (5.49%) | 9 (4.59%) | |
| **How many days has a facemask been used?** | | | | |
| 1 day | 304 (70.21%) | 155 (65.40%) | 149 (76.02%) | 0.121 |
| 2 days | 103 (23.79%) | 65 (27.43%) | 38 (19.93%) | |
| 3 days | 14 (3.23%) | 9 (3.80%) | 5 (2.55%) | |
| > 4 days | 12 (2.77%) | 8 (3.38%) | 4 (2.04%) | |
| **Period use** | | | | |
| < 1 h | 2 (0.46%) | 2 (0.84%) | 0 (0.00%) | 0.009 |
| 1–2 h | 47 (10.85%) | 26 (10.97%) | 21 (10.71%) | |
| 3–4 h | 197 (45.50%) | 122 (51.48%) | 75 (38.27%) | |
| > 4 h | 187 (43.19%) | 87 (36.71%) | 100 (51.02%) | |

students understood more about facemask disposal than the NHS students ($p < 0.05$; Table 2). The knowledge levels between the male and female students were significantly different, even though there were no variations in academic grade. In this study, more than 90% of the students correctly answered the questions regarding the appropriate facemask disposal containers (Table 3). The HS students demonstrated a greater understanding of the management of used facemasks and were better able to differentiate the requirements for facemask waste management versus general solid waste management than the NHS students ($p < 0.05$). However, up to 70% of students were confused regarding the proper disposal of used facemasks. They thought used facemasks could be treated in the same way as general waste. Furthermore, less than half the students recognized the best way to dispose of used facemasks.

## Facemask disposal practices

The students demonstrated good levels of facemask disposal practices, as evidenced by the average score of 12.08 ± 3.06. The HS students exhibited good practices with regard to facemask disposal, but the practices of the NHS students were poor. There were significant differences in practice between the students based on gender, academic program, and grade

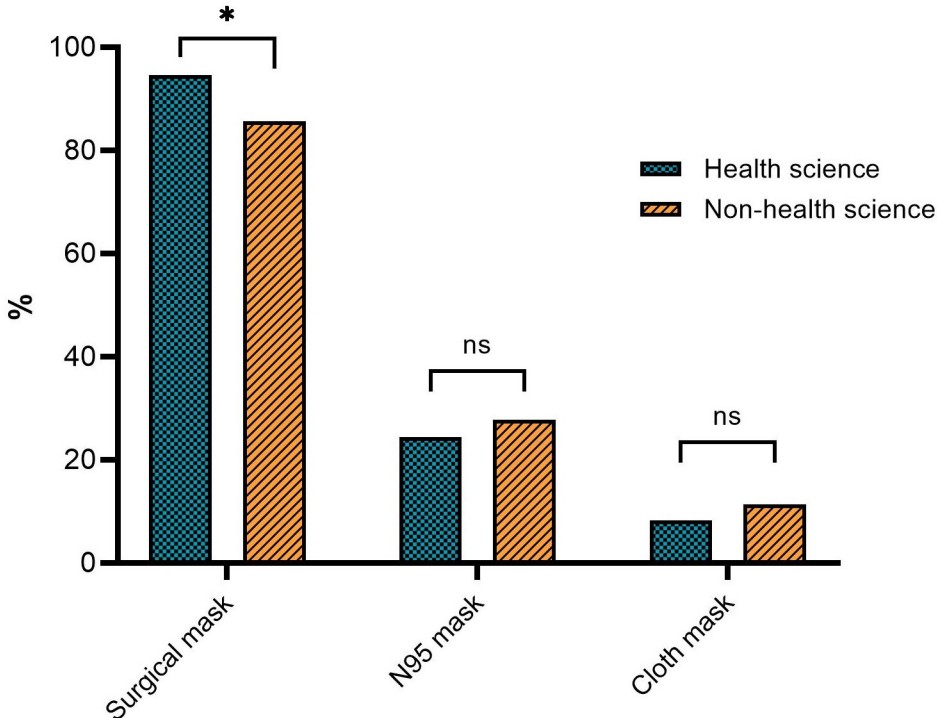

**Fig 1. The type of facemasks usage among HS (N = 196) and NHS students (N = 237).** * indicates a significant different between HS and NHS students at $p < 0.05$, while ns indicates no significant.

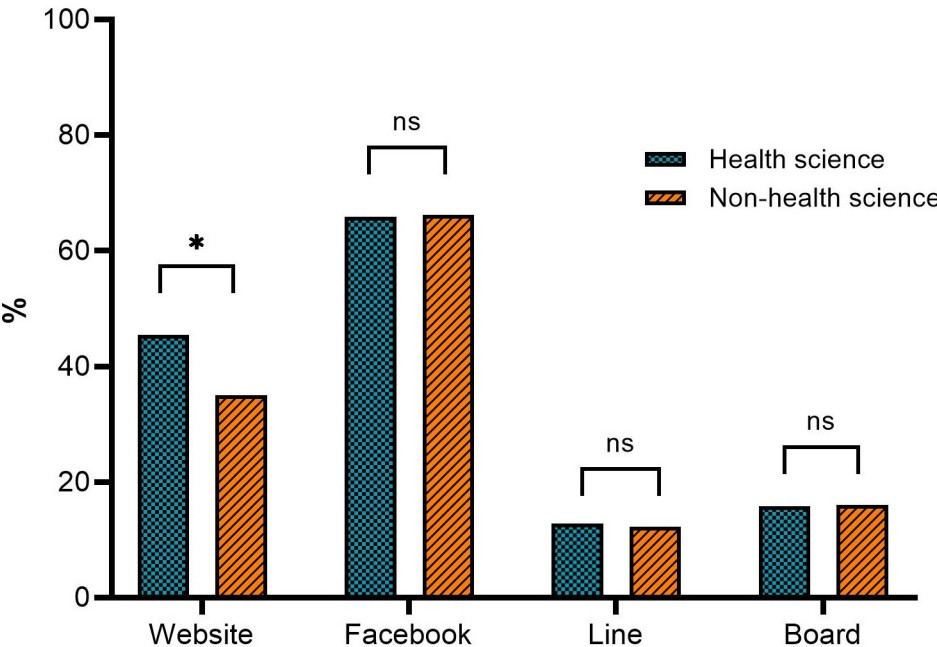

**Fig 2. The information sources of facemask disposal among HS (N = 196) and NHS students (N = 237).** * indicates a significant different between HS and NHS students at $p < 0.05$, while ns indicates no significant.

**Table 2. Responses to the questions related to facemask disposal knowledge and practices.**

| Variables | Knowledge level | | | Practice level | | |
|---|---|---|---|---|---|---|
| | Poor n (%) | Good n (%) | *p*-value | Poor n (%) | Good n (%) | *p*-value |
| **Sex** | | | | | | |
| Male | 51.47% | 48.53% | 0.050 | 63.24% | 36.76% | 0.005 |
| Female | 38.36% | 61.61% | | 44.11% | 55.89% | |
| **Academic program** | | | | | | |
| NHS | 44.73% | 55.27% | 0.049 | 53.59% | 46.41% | 0.021 |
| HS | 35.20% | 64.80% | | 39.29% | 60.71% | |
| **Academic grade** | | | | | | |
| 1st year | 41.67% | 58.33% | 0.259 | 52.78% | 47.22% | 0.004 |
| 2nd year | 46.25% | 53.75% | | 43.75% | 56.25% | |
| 3rd year | 34.38% | 65.63% | | 39.06% | 60.94% | |
| 4th year | 29.73% | 70.27% | | 29.73% | 70.27% | |

($p < 0.05$; Table 2). When their academic grades were compared, the senior students achieved the highest average score (12.78 ± 3.37) for practice. Notwithstanding, no significant differences were observed between the HS and NHS students with regard to facemask disposal practices.

More than 90% of the students demonstrated appropriate practices before throwing their facemasks away and washed their hands after discarding the masks to limit viral transmission (Table 4). Only a few students discarded their used facemasks in the correct trash bins. More than half the students collected used facemasks in plastic PET bottles, while 85% collected them in a plastic or zip-lock bags before disposing of them.

**Table 3. Knowledge of used facemask disposal among the HS and NHS students (*N* = 433).**

| Question | Academic program | Response, Number (%) | | *p* |
|---|---|---|---|---|
| | | True | Fales | |
| K1 Infectious waste means all waste generated by hospitals or healthcare facilities only. | NHS | 95 (40.08%) | 142 (59.92%) | 0.922 |
| | HS | 80 (40.82%) | 116 (59.18%) | |
| K2 Used facemasks for general people, who are not sick and healthy, are not considered infectious waste. | NHS | 95 (40.08%) | 142(59.92%) | 0.032 |
| | HS | 69 (35.20%) | 127 (64.80%) | |
| K3 Used face masks must be collected in closed containers. | NHS | 215 (90.72%) | 22 (9.28%) | 0.202 |
| | HS | 185 (94.39%) | 11 (5.61%) | |
| K4 Used facemasks for general people is not sick and healthy can be discarded in with general waste. | NHS | 88 (37.13%) | 149 (62.87%) | <0.001 |
| | HS | 41 (20.92%) | 155 (79.08%) | |
| K5 Trash bins for disposed face masks should be clearly labeled infectious waste. | NHS | 221 (93.25%) | 16 (6.75%) | 0.545 |
| | HS | 186 (94.90%) | 10 (5.10%) | |
| K6 The used facemasks can be treated the same as general solid waste. | NHS | 79 (33.33%) | 158 (66.67%) | 0.026 |
| | HS | 46 (23.47%) | 150 (76.53%) | |
| K7 Used facemask should be cut into small parts before discarding. | NHS | 166 (70.04%) | 71 (29.96%) | 0.834 |
| | HS | 135 (68.88%) | 61 (31.12%) | |
| K8 The best way to dispose of the used facemask is incineration. | NHS | 115 (48.52%) | 122 (51.48%) | 0.289 |
| | HS | 85 (43.37%) | 111 (56.63%) | |
| K9 The best way to dispose of the discarded mask is in landfill. | NHS | 102 (43.04%) | 135 (56.96%) | 0.239 |
| | HS | 73 (37.24%) | 123 (62.76%) | |
| K10 Used face masks must be discarded in a red bin designated specifically for infectious waste. | NHS | 216 (91.14%) | 21 (8.86%) | 0.363 |
| | HS | 184 (93.88%) | 12 (6.12%) | |

**Table 4. Facemask disposal practices among the HS and NHS students (N = 433).**

| Question (correct answer) | Academic program | Response, Number (%) | | | p |
|---|---|---|---|---|---|
| | | Never | Sometime | Always | |
| P1 I reuse disposable facemask to reduce the amount of waste. | NHS | 110 (46.41%) | 115 (48.52%) | 12 (5.06%) | 0.163 |
| | HS | 109 (55.61%) | 79 (40.31%) | 8 (4.08%) | |
| P2 I always fold used facemasks before discarding them to reduce the risk of spreading germs to others. | NHS | 20 (8.44%) | 123 (51.90%) | 94 (39.66%) | 0.149 |
| | HS | 15 (7.65%) | 85 (43.37%) | 96 (48.98%) | |
| P3 I collect used facemasks in plastic PET bottles to prevent the spread of germs in air. | NHS | 107 (45.15%) | 101 (42.62%) | 29 (12.24%) | 0.589 |
| | HS | 95 (48.47%) | 74 (37.76%) | 27 (13.87%) | |
| P4 I collect used facemasks in a plastic bag or zip-lock bag before discarding them in the garbage bag. | NHS | 42 (17.72%) | 118 (49.79%) | 77 (32.49%) | 0.120 |
| | HS | 22 (11.22%) | 98 (50.00%) | 76 (38.78%) | |
| P5 I discard used facemask mixed with general solid wastes. | NHS | 68 (28.69%) | 129 (54.43%) | 40 (16.88%) | 0.463 |
| | HS | 53 (27.04%) | 117 (59.69%) | 26 (13.27%) | |
| P6 I discard used facemask in a plastic bag, tie and write "Infectious waste" on the bag. | NHS | 85 (35.86%) | 115 (48.52%) | 37 (15.61%) | 0.215 |
| | HS | 61 (31.12%) | 92 (46.94%) | 43 (21.94%) | |
| P7 I dispose used face masks in red bin designated specifically for infectious waste. | NHS | 30 (12.66%) | 130 (54.85%) | 77 (32.49%) | 0.102 |
| | HS | 15 (7.65%) | 102 (52.04%) | 79 (40.31%) | |
| P8 I discarded the used facemask without consideration of trash bin categories. | NHS | 70 (29.54%) | 137 (57.81%) | 30 (12.66%) | 0.587 |
| | HS | 67 (34.18%) | 107 (54.59%) | 22 (11.22%) | |
| P9 I set up special trash cans/bins for disposed masks in residence. | NHS | 67 (28.27%) | 129 (54.43%) | 41 (17.30%) | 0.729 |
| | HS | 49 (25.00%) | 110 (56.12%) | 37 (18.88%) | |
| P10 I wash my hands after taking off and disposing of the mask. | NHS | 22 (9.28%) | 118 (49.79%) | 97 (40.93%) | 0.054 |
| | HS | 14 (7.14%) | 79 (40.31%) | 103 (52.55%) | |

## Relationship between knowledge and practice

The results of the correlation between knowledge and practices and demographic characteristics are shown in Table 5. The knowledge on facemask disposal was significantly related to sex ($r = 0.097$, $p < 0.05$) and academic program ($r = 0.097$, $p < 0.05$), as revealed by the obtained

**Table 5. Correlation between facemask disposal knowledge and practices.**

| Variables | Knowledge | | | | Practice | | | |
|---|---|---|---|---|---|---|---|---|
| | r | Sig | r | Sig | r | Sig | r | Sig |
| Sex | 0.097* | 0.043 | 0.147** | 0.002 | 0.139** | 0.004 | 0.147 | 0.002 |
| Academic grade | 0.054 | 0.259 | | | 0.147** | 0.002 | | |
| Academic program | 0.097* | 0.045 | | | 0.143** | 0.003 | | |

**Table 6. Factors associated with facemask disposal knowledge and practices (logistic regression).**

| Variables | Knowledge level | | | Practice level | | |
|---|---|---|---|---|---|---|
| | Crude OR[a] (95% CI) | Adjusted OR[a] (95% CI) | *p*-value | Crude OR[a] (95% CI) | Adjusted OR[a] (95% CI) | *p*-value |
| Sex | | | | | | |
| Male | 0.587 (0.349–0.987) | 0.691 (0.400–1.194) | 0.186 | 0.459 (0.269–0.783) | 0.469 (0.267–0.825) | 0.009 |
| Female | 1 | 1 | | 1 | 1 | |
| Academic grade | | | | | | |
| 1st year | 0.592 (0.280–1.252) | 0.709 (0.322–1.559) | 0.392 | 0.379 (0.179–0.799) | 0.427 (0.193–0.948) | 0.036 |
| 2nd year | 0.492 (0.214–1.128) | 0.593 (0.250–1.408) | 0.236 | 0.544 (0.237–1.250) | 0.724 (0.300–1.750) | 0.474 |
| 3rd year | 0.808 (0.337–1.935) | 0.847 (0.347–2.065) | 0.715 | 0.660 (0.278–1.568) | 0.652 (0.266–1.597) | 0.349 |
| 4th year | 1 | 1 | | 1 | 1 | |
| Academic program | | | | | | |
| Non- health science | 0.671 (0.455–0.991) | 0.812 (0.534–1.235) | 0.329 | 0.560 (0.382–0.823) | 0.708 (0.468–1.073) | 0.104 |
| Health science | 1 | 1 | | 1 | 1 | |
| Knowledge level | | | | | | |
| Poor | | | | 0.548 (0.372–0.808) | 0.594 (0.399–0.886) | 0.011 |
| Good | | | | 1 | 1 | |
| Practice level | | | | | | |
| Poor | 0.548 (0.372–0.808) | 0.595 (0.399–0.887) | 0.011 | | | |
| Good | 1 | 1 | | | | |

OR: Odds ratio.

[a]OR for included explanatory factors: adjusted with sex, academic grade, and academic program, Knowledge, and Practice

Reference categories: 1 and better or more appropriate

Respondents scoring higher than the median were assessed as having "good knowledge" and "good practices" regarding facemask disposal.

coefficients of correlation. Facemask disposal practices also showed a correlation with sex ($r = 0.139$, $p < 0.01$), academic grade ($r = 0.147$, $p < 0.01$), and academic program ($r = 0.143$, $p < 0.01$). Furthermore, there was a positive correlation between facemask disposal knowledge and practices ($r = 0.147$, $p < 0.01$).

## Factors associated with facemask disposal knowledge and practices

After adjusting for other variables in the multivariable logistic regression analysis (Table 6), only poor facemask disposal practices was found to be significantly associated with knowledge regarding facemask disposal ($p < 0.05$). The students with poor facemask disposal practices were 40.50% less likely to have good knowledge than those with good practices (AOR: 0.595; 95% CI: 0.399–0.887).

Fourth-year students, females, and students with good knowledge were found to be significantly associated with good facemask disposal practices ($p < 0.05$). The fourth-year students were 57.30% less likely to have poor facemask disposal practices than the first-year students (AOR: 0.427; 95% CI: 0.193–0.948), and the female students were 53.10% less likely to have poor practices than the male students (AOR: 0.469; 95% CI: 0.267–0.825). Furthermore, the students with good knowledge about facemask disposal were 40.60% less likely to demonstrate poor practices than the students with poor facemask disposal knowledge (AOR: 0.594; 95% CI: 0.399–0.886).

## Discussion

Wearing a facemask is the most effective intervention for preventing COVID-19 infection and controlling the spread of the virus [8, 9]. Even though the level of contagion has declined,

people in Thailand are still required to wear a facemask in public. Similarly, according to university policy, students at university are required to wear a facemask during class. Millions of used facemasks are therefore disposed of as waste, which may lead to a severe environmental problem.

In this study, we determined the facemask disposal knowledge and practices of 433 university students in Thailand. Our results showed that surgical masks are the most common type of mask used by students, followed by N95 masks. Most of the students (83.37%) wore a different facemask one to two times daily and never reused them (70.21%). Studies in India (79.30%) and Italy (64.90%) have similarly reported a high prevalence of daily facemask changing [17, 32]. However, reuse was also observed for all facemask types, particularly surgical masks. A previous study indicated that surgical masks are only used once [33]. In our study, up to 45.50% of the students wore a single-use facemask for three to four hours, while 43.19% wore one for more than four hours. A study showed that 42.27% of patients in a dermatology clinic wore single-use facemasks for only one to three hours at a time [32], and most healthcare workers in non-COVID areas wore such masks for only one to four hours [34]. The long-term use of a surgical mask reduces its filtration effectiveness, as filtration effectiveness decreases after four hours of wearing time [35].

In terms of information sources, Facebook was the most common platform for acquiring information about facemask disposal among the students. The HS students obtained more information about facemask disposal from government agency websites than the NHS students. Our findings indicated that various platforms could be used to provide information about facemask disposal, which is in line with a previous study on the use of various forms of communication to enhance students' knowledge [36].

Nevertheless, the overall knowledge of facemask disposal among the university students in our study was poor, which indicates a lack of information regarding the proper disposal of used facemasks. In almost all the knowledge questions, the HS students achieved a higher percentage of correct answers than the NHS students. Our findings thus showed that the HS students had a better understanding of infectious waste identification and management than the NHS students. Most of the students had good knowledge of the appropriate facemask disposal containers, namely, that trash bins for the disposal of facemasks are clearly labeled "infectious waste," facemasks must be discarded in a red bin, and facemasks must be collected in closed containers. The students attained 94.00%, 92.38%, 92.38%, respectively, for these questions. Our findings were consistent with those from a study conducted with healthcare workers in clinics in Thailand, which found that 96.5% of the respondents used red color coding to identify infectious waste [23]. However, 60%–70% of the students in our study still needed to understand the sources of infectious waste and the definition, separation, and management of infectious waste, particularly facemasks. Several studies have recommended cutting facemasks into small pieces before discarding them to avoid reuse. Because used facemasks are infectious waste, cutting them into small pieces may increase the volumes of infectious waste in the environment. This practice may also complicate waste disposal and have many consequences, particularly for humans and the environment.

Our study's findings revealed a weakly positive correlation between knowledge of facemask disposal and the variables sex and academic program. In the Philippines, a correlation was also found between sex and knowledge of single-use facemask disposable among students [25]. The researchers determined that the male students in their study had a higher level of understanding than their female counterparts, and the participants with high levels of knowledge were identified as young males with a high level of educational attainment. The correlation between academic programs and knowledge levels has also been observed in a study in Ethiopia [37]. The researchers found that students from the colleges of natural and computational sciences

and social science and law had significantly lower knowledge of facemask utilization than those from the colleges of medicine and health science. We found no significant difference in knowledge levels across the academic year, which indicated that students at different levels received the information equally. In contrast, a study in the Philippines reported that education level significantly influenced knowledge of single-use facemask disposal [25].

According to the multivariable logistic regression analysis in our study, no factors influenced the students' knowledge of facemask disposal. This could have been the result of inadequate classroom teaching on facemask disposal and the poor quality of the information obtained from online sources. Proper facemask disposal was essential for preventing negative health and environmental effects during the pandemic. Tuition on the proper use and disposal of facemasks, along with sustained facemask disposal, should therefore be provided to students to reduce future health and environmental problems.

All the students in our study exhibited good practices regarding facemask disposal. No significant differences were noted between the HS and NHS students with respect to facemask disposal practices. Half the students had never reused disposable facemasks to reduce waste volumes, while others had only done so on occasion. Our findings indicated that university students were more concerned about their health during the pandemic than environmental issues. The students tried to prevent viral transmission by folding their used facemasks (95.38%), discarding the used facemasks in plastic bags and tying them (66.28%), and collecting used facemasks in zip-lock bags (85.22%) before throwing them away. Similar percentages of students in the Philippines collected used facemasks in closed containers, such as zip-lock and tied plastic bags [25]. In Asian countries, it is recommended to fold used facemasks inward before discarding them in closed containers to prevent viral transmission and reuse by others [13, 38]. The Indonesian government has also recommended that facemasks be disinfected before throwing them away.

Due to facemasks being infectious waste that requires a specific disposal method, the way they are discarded is important. Facemasks must be collected separately from other waste using a specific bin. This bin needs to be disinfected, and transportation to an incineration facility must be undertaken by specially trained personnel with proper PPE and special vehicles [13]. Facemasks should be disposed of by incineration at a sufficiently high temperature (800˚C – 1200˚C) to kill microorganisms and reduce the volume and mass of such waste. Two main thermal technologies used to dispose of facemasks and other COVID-19 waste are the high-temperature pyrolysis technique and the medium-temperature microwave technique [39]. By way of contrast, municipal solid waste should be dumped in open fields without pretreatment [40–42]. Roughly 30% of the students in our study were cognizant of the need to discard their facemasks in the appropriate trash bins and indeed did so. These findings are consistent with a study conducted in the Philippines, which found that 31.70% of the general public never throw their facemasks away in recycling bins [25]. However, up to 70.00% of the students in our study discarded their facemasks in general solid waste bins without regard for the bin type. The same was found in Italy, with 70.50% of patients discarding their facemasks in general waste bins [32]. In this study, the percentage of students who discarded their used facemasks as mixed waste was two times higher than in a previous survey [18]. The improper disposal of used facemasks can cause serious environmental problems due to the use of non-degradable materials and health problems related to pathogens. As revealed in this study, infectious waste disposal was misunderstood, and this could have contributed to the improper practice of discarding facemask waste in general solid waste bins. A previous study has also described the improper disposal of medical waste in general waste bins due to a lack of knowledge [23].

In this study, the students discarded their facemasks in plastic PET bottles to prevent disease transmission. However, in terms of environmental management, this approach needs to

be improved. Plastic PET bottles are recyclable waste that can be reused; however, it becomes infectious waste when it comes into contact with a facemask. Although such infectious waste must be disposed of where incineration is used, this generates air pollution [43]. Furthermore, viral spread could occur after waste sorters remove facemasks from plastic PET bottles so that the bottles can be sold. Content regarding facemask disposal in PET bottles is widely shared on social media in Thailand to prevent disease transmission in households. This unsatisfactory practice could thus be the result of information shared on social media, as this was presented as the primary source of information in this study.

Our findings revealed a positive correlation between demographic characteristics (the variables sex, academic grade, and academic program) and facemask disposal knowledge and practices. The female students in our study had better practices than the male students, and the HS students exhibited better practices than the NHS students. Proper facemask disposal has become increasingly the norm as the students have acquired more educational experience. The students in the HS-related majors and those in a senior academic year were able to apply their knowledge regarding facemask disposal more appropriately due to their academic curriculum experience and scientific knowledge. A study of facemask use among Vietnamese students found a significant correlation between facemask disposal practices and sex, as well as academic major [44]. In the Philippines, the practice of single-use mask disposal also correlated with sex, academic grade, and knowledge level [25]. Furthermore, a study in Indonesia demonstrated the correlation between undergraduate medical students' knowledge and attitudes and their practices with respect to COVID-19 [45]. The results emphasized the importance of HS students: as future health professionals and public educators, they will play a critical role in helping prevent and control disease in communities during pandemics [46].

Notably, the results of our study indicated the factors associated with facemask disposal practices among students. Using the multivariate analysis, we determined that sex, academic grade, and knowledge level influenced students' facemask disposal practices. Senior students demonstrated better facemask disposal practices than first-year students, whereas students with good knowledge demonstrated better practices than others. In contrast, a study in Ethiopia on facemask use revealed no association between knowledge and practices, and academic programs had no influence on facemask disposal practices [37].

Our study had some limitations. First, we only examined the students' facemask disposal knowledge and practices, so their attitudes toward facemask disposal were not defined. Research on attitudes should be performed in future studies to more fully describe the factors influencing the facemask disposal practices of the study group. The second limitation was the proportions of students in the sample group evaluated to ascertain the influence of the variables sex and academic grade on facemask disposal. Because of the restrictions on sample access during the COVID-19 pandemic, simple random sampling was used to collect our data. The sample in this study thus overly represented female and first-year students. A more systematic and comprehensive sampling approach is thus required to improve the representativeness of the findings. Another limitation of the current study was the possibility of the participants giving socially acceptable responses. Because this study relied on self-reported data, it is possible that the participants responded positively to the practice questions based on what they assumed would be expected of them [47].

## Recommendations

Our study findings revealed the influence of knowledge on proper facemask disposal practices, so information on infectious waste management should be promoted more extensively. In addition, university students' awareness of facemask disposal methods should be increased by

continuously educating them via the most relevant social media platforms with the aim of improving the effectiveness of their waste management methods and to reduce the risks posed by inappropriate facemask disposal to human health and the environment [48–50]. Previous studies have demonstrated that environmental awareness and knowledge are essential for participation in recycling programs [51]. Moreover, subjective norms, perceived behaviors, moral obligations, situational factors, and motivations could explain students' intentions and behaviors regarding waste separation at source and thereby contribute to the promotion of pro-environmental intentions and behaviors [52–54]. Our recommendations may be helpful in promoting proper facemask disposal practices among students if the theory of planned behavior is followed. This theory assumes that several reasons or constructs, including attitudes, subjective norms, and perceived behavioral control, can be engaged to form intentions to perform a specific behavior [55]. Furthermore, a lack of the necessary infrastructure is a significant obstacle for recycling; thus, having sufficient available bins is a significant consideration when engaging students in the source separation of facemasks for disposal [56, 57].

## Conclusion

The current study determined the facemask disposal knowledge and practices among university students in Thailand. The results of this study revealed the students' poor levels of understanding regarding facemask disposal, and their practices could have been better. Additionally, demographic factors did not influence the students' knowledge of facemask disposal. Our findings highlight the influence of facemask disposal knowledge on students' disposal practices, and the correct knowledge should be promoted more extensively. Furthermore, continuous reinforcement by raising awareness and educating students on proper facemask disposal combined with adequate infectious waste disposal facilities could help reduce the environmental contamination of infectious waste while also enhancing proper general waste management. Finally, the outcomes of proper infectious waste management may also result in reduced restrictions in future pandemics.

## Author Contributions

**Conceptualization:** Narisara Kaewchutima, Nopadol Precha.

**Data curation:** Narisara Kaewchutima, Nopadol Precha, Netnapa Duangkong, Anthika Jitbanjong.

**Formal analysis:** Narisara Kaewchutima, Nopadol Precha, Ni Made Utami Dwipayanti.

**Methodology:** Narisara Kaewchutima, Nopadol Precha.

**Supervision:** Narisara Kaewchutima.

**Visualization:** Narisara Kaewchutima, Nopadol Precha.

**Writing – original draft:** Narisara Kaewchutima, Nopadol Precha, Ni Made Utami Dwipayanti.

**Writing – review & editing:** Narisara Kaewchutima, Nopadol Precha, Ni Made Utami Dwipayanti.

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
