## [Decision Letter · Decision Letter 0]

6 Mar 2023

PONE-D-23-04502Knowledge and practice of facemask disposal among university students in Thailand:a new normal post the COVID-19 pandemicPLOS ONE

Dear Dr. KAEWCHUTIMA,

Thank you for submitting your manuscript to PLOS ONE. After careful consideration, we feel that it has merit but does not fully meet PLOS ONE’s publication criteria as it currently stands. Therefore, we invite you to submit a revised version of the manuscript that addresses the points raised during the review process.

Moreover, reviewers are critically vocal to;Improve language of manuscript by native with same background.Add outcome of study as separate section.Add a short questionnaire to find out reasons behind such lacking awareness among students.And to indicate why fame students were more aware compared to female students.

We look forward to receiving your revised manuscript.

Kind regards,

Sadia Ilyas, Ph.D.

Academic Editor

PLOS ONE

Journal Requirements:

a) Did participants provide their written or verbal informed consent to participate in this study?

Additional Editor Comments:

Reviewer-1

The authors present an interesting piece providing data and perspective on knowledge and perspective of mask disposal among university students. The data gathering and statistical analysis has been rigorously done, as well as the discussion part. Several actions might be done to improve the paper more:

1. Comprehensive proofreading by native English speakers with relevant backgrounds might improve the paper (if this has not been done).

2. If considered relevant, the authors could present more perspective of knowledge and practice among university students during the pandemic from the more or less same demographic (in this case, e.g., Southeast Asia). These following papers could be considered to be cited: Adli I et al (2022) http://dx.doi.org/10.1371/journal.pone.0262827 ; Lazarus G et al (2021) https://bmcmededuc.biomedcentral.com/articles/10.1186/s12909-021-02576-0

On a side note, I support the authors in disseminating the paper more, such as by panel discussion, webinar, etc., involving central and local policymakers, local academicians, youth representatives, etc.

Reviewer-2

In the present manuscript, the authors performed an investigation on the knowledge and practice of facemask disposal among university students in Thailand. The study is related to COVID-waste management, hence, it is timely and of interests. They used SPSS 25 to analyze the data obtained by the survey conducted within the university. The study is well conducted, however, this reviewer has certain points to mention before its final consideration.

1). My major concern is that after a hilarious pandemic like COVID-19, if up to 70% of students are confused regarding the proper management of used facemasks, it is a serious matter of concern. My point is that why the authors did not include the questionnaires on the topic to find out the reasons behind such lacking awareness among the students. It is expected that the younger generation should be more aware of that. Add a note and additional data.

2). Line 299-300: The authors should mention that why "male have a greater level of understanding than female"

3). The authors should add a separate recommendation section to overcome the issues find out by this study.

4). The authors should include some more articles on discussion like https://doi.org/10.1016/j.scitotenv.2020.141652

5). The authors should add the possible waste management techniques related to face masks.

Reviewers' comments:

Reviewer's Responses to Questions

**Comments to the Author**

1. Is the manuscript technically sound, and do the data support the conclusions?

Reviewer #1: Yes

Reviewer #2: Yes

2. Has the statistical analysis been performed appropriately and rigorously? 

Reviewer #1: Yes

Reviewer #2: Yes

3. Have the authors made all data underlying the findings in their manuscript fully available?

Reviewer #1: Yes

Reviewer #2: Yes

4. Is the manuscript presented in an intelligible fashion and written in standard English?

Reviewer #1: Yes

Reviewer #2: Yes

5. Review Comments to the Author

Reviewer #1: The authors present an interesting piece providing data and perspective on knowledge and perspective of mask disposal among university students. The data gathering and statistical analysis has been rigorously done, as well as the discussion part. Several actions might be done to improve the paper more:

1. Comprehensive proofreading by native English speakers with relevant backgrounds might improve the paper (if this has not been done).

2. If considered relevant, the authors could present more perspective of knowledge and practice among university students during the pandemic from the more or less same demographic (in this case, e.g., Southeast Asia). These following papers could be considered to be cited: Adli I et al (2022) http://dx.doi.org/10.1371/journal.pone.0262827 ; Lazarus G et al (2021) https://bmcmededuc.biomedcentral.com/articles/10.1186/s12909-021-02576-0

On a side note, I support the authors in disseminating the paper more, such as by panel discussion, webinar, etc., involving central and local policymakers, local academicians, youth representatives, etc.

Regards,

Reviewer

Reviewer #2: In the present manuscript, the authors performed an investigation on the knowledge and practice of facemask disposal among university students in Thailand. The study is related to COVID-waste management, hence, it is timely and of interests. They used SPSS 25 to analyze the data obtained by the survey conducted within the university. The study is well conducted, however, this reviewer has certain points to mention before its final consideration.

1). My major concern is that after a hilarious pandemic like COVID-19, if up to 70% of students are confused regarding the proper management of used facemasks, it is a serious matter of concern. My point is that why the authors did not include the questionnaires on the topic to find out the reasons behind such lacking awareness among the students. It is expected that the younger generation should be more aware of that. Add a note and additional data.

2). Line 299-300: The authors should mention that why "male have a greater level of understanding than female"

3). The authors should add a separate recommendation section to overcome the issues find out by this study.

4). The authors should include some more articles on discussion like https://doi.org/10.1016/j.scitotenv.2020.141652

5). The authors should add the possible waste management techniques related to face masks.

6. PLOS authors have the option to publish the peer review history of their article (what does this mean?). If published, this will include your full peer review and any attached files.

Reviewer #1: **Yes: **Nico Gamalliel, MD

Reviewer #2: No

---

## [Author Response · Author response to Decision Letter 0]

29 Mar 2023

Reviewer 1: 

1. The manuscript has been proven by native English speaker as attached certificate.

2. Thank you very much for the great suggestions and references for discussion. We added more references on the discussion part to show the important role of the students in controlling and preventing the diseases during pandemic.

3. Thank you very much for your kind suggestion. We have planned to share our findings to the university and local policymakers. Also, we had shared this study in the international symposium of waste management.

Reviewer 2:

1. The authors appreciate this critical point made by reviewer #2. We agree that this is important to be addressed.

It is also an interesting issue for us about poor knowledge on facemask disposal among the students. In addition, we are afraid we cannot add the questionnaire and additional data on awareness in the original sample group. Therefore, we will disseminate the findings of this study to the university’s policymaker to initiate the educational programs of infectious waste management in order to increase the awareness and proper practice in the students as mentioned in the recommendation.

2. Thank you very much for your comment. According to your comment, it is the results from the previous study that we used in discussion. However, we added the reasons why male have the greater level of understanding than female for more perfect discussion.

3. Thank you very much, we have separated the recommendation into the new section. 

4. Thank you very much for the useful reference. We added more reference about incineration of used facemask in the discussion.

5. Thank you very much for useful suggestions on the additional management techniques related to facemask. We added possible waste management techniques in the discussion.

---

## [Editor Report · Decision Letter 1]

3 Apr 2023

Knowledge and practice of facemask disposal among university students in Thailand: A new normal post the COVID-19 pandemic

PONE-D-23-04502R1

Dear Dr. Narisara,

We’re pleased to inform you that your manuscript has been judged scientifically suitable for publication and will be formally accepted for publication once it meets all outstanding technical requirements.

Kind regards,

Sadia Ilyas, Ph.D.

Academic Editor

PLOS ONE
---

## [Editor Report · Acceptance letter]

4 Apr 2023

PONE-D-23-04502R1 

Knowledge and practice of facemask disposal among university students in Thailand: A new normal post the COVID-19 pandemic 

Dear Dr. Kaewchutima:

I'm pleased to inform you that your manuscript has been deemed suitable for publication in PLOS ONE. Congratulations! Your manuscript is now with our production department. 

Kind regards, 

on behalf of

Prof. Sadia Ilyas 

Academic Editor

PLOS ONE